# The Short- and Long-Term Impacts of Hurricane Irma on Florida Agricultural Leaders as Early Emergency Responders: The Importance of Workplace Stability

**DOI:** 10.3390/ijerph17031050

**Published:** 2020-02-07

**Authors:** Lynn M. Grattan, Angela Lindsay, Yuanyuan Liang, Kelsey A. Kilmon, Scott Cohen, Tracy Irani, John Glenn Morris

**Affiliations:** 1Department of Neurology, Neuropsychology Program, University of Maryland School of Medicine, Baltimore, MD 21201, USA; Kelsey.Roberts@som.umaryland.edu; 2Department of Family, Youth, and Community Sciences, College of Agricultural and Life Sciences, University of Florida, Gainesville, FL 32611, USA; ablindsey@ufl.edu (A.L.); Irani@ufl.edu (T.I.); 3Department of Epidemiology and Public Health, Biostatistics and Informatics, University of Maryland School of Medicine, Baltimore, MD 21201, USA; yliang@som.umaryland.edu; 4Department of Emergency Medicine, College of Medicine, University of Florida, Gainesville, FL 32611, USA; scohen211@ufl.edu; 5Department of Medicine, College of Medicine, FL University of Florida, Gainesville, FL 32611, USA; jgmorris@epi.ufl.edu

**Keywords:** hurricanes, disasters, non-professional first responders, Hurricane Impact, agricultural extension workers, workplace stability, post-disaster behavioral health

## Abstract

The impacts of hurricane-related disasters in agricultural communities include extensive losses of fields, orchards, and livestock, the recovery of which could span many years. Agricultural Extension Agents (EAs) try to manage and mitigate these losses, while simultaneously overseeing emergency shelter operations. These non-professional emergency responders face numerous potential stressors, the outcomes of which are minimally known. This study examined the short- and long-term medical and behavioral outcome of 36 University of Florida Agricultural Extension Agents within two months and one year after Hurricane Irma, Florida, USA, taking into consideration personal/home and work-related hurricane impacts. Regression analyses indicated that combined home and work hurricane impacts were associated with greater anxiety, depression, and medical symptoms controlling for age and number of prior hurricane experiences within two months of landfall. One year later, depression symptoms increased as well as the use of negative disengagement coping strategies for which stability of the work environment was protective. The findings suggest that advanced training in emergency response, organization and time management skills, time off and temporary replacement for personally impacted EAs, and workplace stability, including enhanced continuity of operations plans, represent critical elements of health prevention and early intervention for this occupational group.

## 1. Introduction

On 10 September 2017, Hurricane Irma made landfall in the Florida Keys as a level four hurricane. Later that same day, Hurricane Irma made landfall again in Southwest Florida as a level three storm. As a result of hurricane-force winds, rain, and flooding, more than 6 million residents were left without power [1] and were forced to deal with structural damage to their homes, property, and workplaces. In the aftermath, there were fuel, food, and construction material shortages, as well as extensive losses of fields, orchards, and livestock. The death toll was 123 [2], and the entire state of Florida was designated a disaster region with the most intensive Federal Emergency Management Agency (FEMA) activity directed to the counties identified in red in Figure 1. These regions represent the core of Florida’s agricultural industry, and the sustained estimated agricultural loss because of Hurricane Irma exceeded $2.5 billion [3]. This includes losses to Florida’s iconic citrus producers as well as sugarcane, avocado, dairy, and beef cattle operations, all of which will likely take years to recover fields, orchards, and livestock. 

The University of Florida Institute for Food and Agricultural Sciences (UF/IFAS) county Extension Agents (EAs) served as early responders in various capacities during and in the aftermath of the storm. They were actively involved in emergency operations and control, leading and managing shelters, operating points of distribution for food and water for residents and livestock, helping to dispose of dead livestock, and providing emotional support to distraught farm owners and workers. Meanwhile, their own homes and workplaces were also impacted.

A wide range of psychological reactivity has been associated with occupational exposures to hurricane-related disasters, with most studies focused on professional emergency responders such as firefighters, police, and ambulance workers. Worldwide, about 10% of these professional responders suffer from Posttraumatic Stress Disorder (PTSD) for up to several years post disaster [4], and after Hurricane Katrina, 25% reported symptoms of depression that persisted for up to 18 months [5]. A wide range of methods have been used to assess these mental health symptoms ranging from clinical judgment to more standardized measures of PTSD and depression, such as the Clinician Administered PTSD Scale and various Posttraumatic Stress Disorder Checklists [6,7], as well as the Center for Epidemiological Studies of Depression Scale, short form (CESD-SF) [8]. Few studies used a formal impact of events scale to assess how the hurricane affected the emergency responders themselves.

Less well known are the impacts upon non-professional emergency service responders, that is, service personnel who are not professional emergency responders but, by virtue of their community-based duties, contribute to the early phases of disaster response [9]. This includes Agricultural Extension Agents as well as others, such as construction and utility workers, public transportation service providers, teachers, faith-based leaders, or public health employees. When the latter group was studied after the 2004 Florida hurricanes (Charley, Frances, Ivan, and Jeanne), 49% of these public health employees reported symptoms of heightened arousal, a component of PTSD (measured by the PTSD Checklist), 9 months post hurricanes [10]. This is higher than the 10% rate of PTSD symptoms for professional emergency responders after Hurricane Katrina using a similar checklist [5]. Moreover, 43% of public health employees reported sleep disturbance on the Patient Health Questionnaire Depression Subscale [11], while 25% of professional early responders reported symptoms of depression after Hurricane Katrina [5] using the CESD [12]. Thus, non-professional emergency responders are at risk of worse mental health outcomes than professional emergency personnel. 

The University of Florida Institute for Food and Agricultural Sciences EAs are at risk of exposure to hurricane-related disasters and poor behavioral health outcomes. This is primarily the result of the large number of hurricanes and hurricane-related disasters in Florida, United States [13]. Hurricane force winds and rain often devastate the agricultural communities served by the EAs due to flooding, crop damage, and livestock deaths. Moreover, the impacts of the hurricane-related disaster are often protracted, requiring the EAs to redirect their strained occupational resources and activities toward disaster recovery for weeks or months. Meanwhile, these EAs typically expand their roles in disaster recovery while managing losses of their own personal resources [14].

### Purpose of Study

The purpose of this study was twofold. First, to characterize the short-term behavioral and physical health outcomes and their correlates for agricultural EAs. Second, to determine the extent to which these acute outcomes were associated with psychological status one year post hurricane, taking into consideration other factors, including hurricane impact at home and in the workplace. It was hypothesized that overall hurricane impact (personal/home and workplace) would be associated with both short- and long-term behavioral health outcomes post hurricane. The ultimate goal was to be able to develop rationally based prevention and early intervention activities for EAs, and potentially other leaders in the agricultural industry, to minimize the distress associated with hurricane-related disasters.

## 2. Materials and Methods

A cross sectional repeated measures study design was used to answer the study questions.

### 2.1. Participants

We invited University of Florida Agricultural Extension Agents (EA’s) from the 27 impacted Florida counties to participate in this study. There are between one and three of these specialized EA’s on staff per impacted county and they were contacted via email and/or telephone two to three weeks post landfall to assess their physical, psychological, and behavioral health status, as well as hurricane impact. Thirty-six male and female UF/IFAS Extension Agents were available for participation during the data collection time period. The response rate was 85%, and participants were examined between three and eight weeks post storm (early recovery phase, T1) with standardized study measures in IFAS facilities. One year post hurricane (T2), all participants were re-contacted by U.S. mail, electronic mail, and telephone. Twenty (55%) of the original participants were available for follow up studies, and data were individually collected at an annual meeting as well as via mail (*n* = 5) for those agents who were unable to attend the meeting. With respect to the participants who did not partake in the follow-up data collection, some changed jobs, while others chose to withdraw from the study for non-identified reasons. There was no overall difference in demographic or outcome measures between those participants who withdrew or remained in the study. Participants were remunerated $25 for participation at each study phase. All research activities were approved by the Institutional Review Board at the University of Maryland, Baltimore (HP-00077599).

### 2.2. Measures

In the first assessment epoch (T1), a study-specific questionnaire with several standardized surveys was administered to assess demographics, health status, number of past hurricane exposures, and current hurricane impact. Specifically, the assessment battery included the Alcohol Use Disorder Identification Test (AUDIT) to assess alcohol use respectively [15]; Coping Orientation to Problems Experienced (COPE) inventory to assess problem-focused, emotion-focused, and disengagement coping styles [16]; Beck Depression Inventory-II (BDI-II) to measure presence and severity of depression symptoms [17]; PTSD Checklist for DSM-5 (PCL-5) to assess anxiety and PTSD [7]; and the Connor Davidson Resilience Scale (CD-RISC) to assess resilience [18]. An adapted Hurricane Impact Scale (EHI) [19,20] was implemented to assess the personal/home, work-related, and total impacts of the storm. The EHI included actual exposure to hurricane force winds, flooding, or injuries (personal impacts); losses of home, personal property, or displacement (home impacts); lost time from work; extra days, hours, and duties at work; and displacement from primary work location. Participant demographic, substance use, and health histories were updated, and psychological and behavioral measures were repeated one year post hurricane (T2). Measures were administered by a psychologist investigator (L.M.G.).

### 2.3. Statistical Methods

Descriptive statistics were used to describe the sample and scores on study measures. Logistical regression modeling was used to identify the variables that were best associated with outcome at Time 1 (T1). One year post hurricane, changes in outcomes (BDI-II, PCL-5, and COPE scale scores) from T1 to T2 were assessed in the same participants (*n* = 20) using a Generalized Estimating Equation (GEE) model in order to account for within-subject correlation due to paired data. The GEE models used the identity link with a correlation structure chosen based on the smallest quasi-likelihood under the independence model criterion. Models were created without and with adjustment for demographics and EHI scores. Multi-collinearity testing was performed using Variable Inflation factors (VIFs) criteria, where covariates with a VIF > 5 were excluded from the adjusted model.

## 3. Results

Demographic information for study participants at both time epochs may be found in Table 1. Overall, the sample consisted of predominantly college educated, Caucasian women, with an income of $75,000 or greater and no children living within the household. This reflects the general demographics of UF/IFAS Extension Agents. Only one of the participants had formal emergency response training and experience.

Alcohol use was minimal, as all participants were within the low-risk range of alcohol problems based upon the AUDIT. The most frequently reported medical symptom complaints at T1 included fatigue (69.4%), irritability (50%), mood changes (38.9%), difficulty concentrating (38.9%), losing or misplacing things (33.3%), sleep disturbance (33.3%), lower back pain (30.6%), and headache (27.8%). The results of the study-specific questionnaires are summarized in Table 2.

During the acute recovery phase (T1), total hurricane impacts (personal, home, and work) were associated with higher scores on the BDI-II and PCL-5 and with the number of medical symptoms reported (Table 3) after controlling for age and number of past hurricane exposures. Further analysis of the T1 data indicated that a wide variety of coping methods were used, with the higher impact group more actively using problem-focused (*M* = 39.59, *SD* = 8.45; *p* = 0.384), emotion-focused (*M* = 42.88, *SD* = 7.97; *p* = 0.564), and disengagement (*M* = 12.47, *SD* = 4.22; *p* = 0.092) strategies than the lower hurricane impact group.

Follow-up data analysis (T2) may be found in Table 4. When assessing change in outcome measures at follow-up, BDI-II, PCL-5, or COPE scale scores were not associated with the time of collection in the unadjusted models. After adjustment for demographics and hurricane impact, BDI-II and COPE disengagement scores significantly increased from the T1 to T2 time periods (Table 4). BDI-II total score increased 3.91 (95% CI: 0.24–7.58) points, and COPE disengagement increased 1.75 (95% CI: 0.40–3.10) points across the study period. In both adjusted models, increased hurricane impact at work measures was significantly associated with an increased BDI-II and COPE disengagement score.

Hurricane-related disasters contribute to considerable distress for early responders, particularly non-professional emergency service personnel. This study found that higher levels of overall Hurricane Irma impact (personal/home and work) were directly associated with more medical, depression, and anxiety symptoms in University of Florida Agricultural Extension Agents during early hurricane recovery. One-year post hurricane, symptoms of depression and the use of disengagement coping strategies increased, with lower work-related hurricane impacts attenuating this outcome. In summary, these findings uniquely suggest that the factors associated with behavioral symptom onset differ from those of persistence or worsening after storm-related disasters in this occupational group, thus partially supporting our hypothesis that overall hurricane impacts (personal/home and work) were associated with short- and long-term outcomes.

## 4. Discussion

Few studies have addressed the short-term impacts of a hurricane-related disaster in non-emergency service professionals. In the present study, new medical symptoms as well as symptoms of anxiety and depression emerged within three months of landfall. With the exception of lower back pain and headache, most of the new medical symptom complaints overlap with symptoms of depression, anxiety, or cognitive inefficiencies (fatigue, sleep disturbance, irritability, and difficulty with concentration and losing or misplacing things). These often occur when stressors outweigh resources for coping [21,22]. Musculoskeletal injuries, such as lower back pain, are considered a common post-hurricane hazard. Headache has been associated with stressful life experiences [23]. The early emergence of these symptoms was associated with the cumulative effects of the hurricane impact rather than disturbance in one domain (personal injury/home or work). This is consistent with resource models of stress and disaster [21,22] whereby the impact of multiple losses in the short term (home, work, financial) trigger a downward spiral of distress. In contrast, one year later, more specific hurricane impacts, i.e., work-related impacts, were associated with behavioral outcome. 

Symptoms of depression and the use of disengagement coping strategies increased over the course of one year post hurricane after controlling for important factors. Unlike the overall hurricane impact being associated with acute psychological reactivity, one year later, work-related hurricane impacts confounded the relationship between depression, disengagement coping strategies, and time. The negative association of work-related impact with time, as well as the positive association with depression and disengagement coping strategies indicate that the lower levels of hurricane impact in the work setting seemed to be protective. That is, the stability of the work environment appears to be protective of persisting or worsening symptoms of depression and the use of disengagement, a maladaptive coping strategy. It is plausible that for professional IFAS Extension Agents, the structure, responsibilities, and social support gained by the stable work environment can mitigate some of the stressors from other life domains (impact at home) and facilitate adaptive behavioral outcomes. 

Study limitations include a relatively small sample size, potential recall bias, and the ability to generalize these findings to other non-professional disaster service providers. We also did not have data on how long the participants were employed as EAs. It is important to note that the UF/IFAS Extension Agents who participated in this study were county employees tenured by the University of Florida. Thus, they had considerable job security and support during the recovery process. These findings may not generalize to agricultural leaders in temporary or transient employment settings. Additionally, while these results could potentially provide significant policy and practical implications for occupational safety and health (OHS), from an international perspective, they would only generalize to regions with similar climactic, socioeconomic, and organizational structures for agricultural leadership. 

### Recommendations

With this in mind, the following implications for prevention and early intervention activities are proposed. First, advanced training for emergency response such as psychological first aid is recommended for these non-professional disaster responders. This should also include advanced organization and time management skills, as well as effective coping skills. Second, having EAs from unaffected counties assist EAs with personal/home damage in the workplace is recommended on a temporary basis. This would allow the impacted EAs the opportunity to manage their personal losses while simultaneously managing the agricultural problems of their communities. Third, the development of continuity of operations plans should be initiated. This includes mechanisms to keep needed resources (generators, fencing, tarps, shelf stable food, water) readily available for transport and distributions as needed. Finally, further research on the impacts on additional occupational groups who are involved in disaster management is needed. 

## 5. Conclusions

Agricultural Extension Agents (EAs) are critical members of emergency hurricane response teams in rural communities. Home and work-related hurricane impacts contribute to poor physical and behavioral outcomes of EAs within the first few weeks of the storm. One year post landfall, however, the stability of the work setting is protective of poor behavioral health outcomes (depression, disengagement coping). The findings highlight the importance of preventive education and planning programs in the workplace to minimize the potential negative outcomes of hurricane-related disasters.

## Figures and Tables

**Figure 1 ijerph-17-01050-f001:**
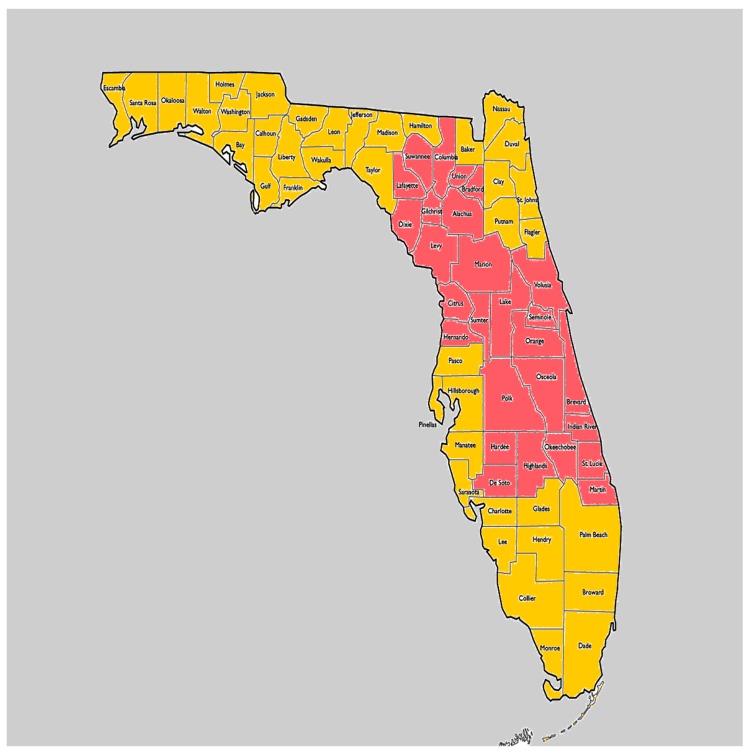
Map highlighting (in red) the Florida (FL) counties identified by FEMA to have the greatest Hurricane Irma related disaster impacts.

**Table 1 ijerph-17-01050-t001:** Demographic and hurricane exposure information for Extension Agents after 3 to 8 weeks (Time 1) and one year (Time 2).

Demographic	Time 1 (n = 20)	Time 2 (n = 20)
**Age—years**		
Mean ± SD	46.50 ± 13.55	48.20 ± 14.78
Min, Max	25, 69	26, 70
**Gender -n (%)**		
Female	24 (66.7)	14 (70.0)
Male	12 (33.3)	6 (30.0)
**Ethnicity-n (%)**		
Hispanic or Latino	3 (8.3)	2 (10.0)
White or Caucasian	32 (88.9)	18 (90.0)
Other	1 (2.8)	0 (0)
**Total Annual Household Income-n (%)**		
$100,000 and Higher	15 (41.7)	11 (55.0)
$15,000 to $49,999	4 (11.2)	3 (15.0)
$50,000 to $74,999	5 (13.9)	2 (10.0)
$75,000 to $99,999	12 (33.3)	4 (20.0)
**Marital Status-n (%)**		
Divorced	1 (2.8)	1 (5.0)
Married	23 (63.9)	15 (75.0)
Partnered	3 (8.3)	2 (10.0)
Single	7 (19.4)	2 (10.0)
Widowed	2 (5.6)	0 (0)
**Children Living in Home-n (%)**		
No	24 (66.7)	14 (70.0)
Yes	12 (33.3)	6 (30.0)
**Years of Education—years**		
Mean ± SD	17.53 ± 1.73	17.80 ± 1.70
Min, Max	13, 20	14, 20
**Number of Endorsed Medical Symptoms**		
Mean ± SD	5.28 ± 5.28	2.40 ± 3.95
Min, Max	0, 22	0, 14
**EHI (Hurricane Irma Exposure) Total Score**		
Mean ± SD	8.33 ± 2.92	N/A
Min, Max	3, 15	

Age, gender, ethnicity, total annual household income, marital status, children living in home, years of education, number of endorsed medical symptoms, and Hurricane Impact Scale (EHI) total score for T1 and T2 are presented in the table above.

**Table 2 ijerph-17-01050-t002:** Descriptive summary of study measures at for Extension Agents after 3 to 8 weeks (Time 1) and one year (Time 2).

	Time 1	Time 2
	(n = 20)	(n = 20)
**Outcomes—mean ± SD**		
BDI-2 Total Score ^1^	6.83 ± 7.41	7.8 ± 5.79
PCL5 ^2^ Total Score	12.56 ± 13.43	10.6 ± 8.65
PCL5 Cluster B Score	3.44 ± 4.46	2.95 ± 2.91
PCL5 Cluster C Sore	1.61 ± 1.96	1.35 ± 1.57
PCL5 Cluster D Sore	3.50 ± 4.15	2.45 ± 3.72
PCL5 Cluster E Sore	4.00 ± 4.28	3.85 ± 3.03
COPE Disengagement	11.47 ± 3.66	12.05 ± 3.20
**Predictors—mean ± SD**		
EHI ^3^ Total Score	8.33 ± 2.92	3.55 ± 3.46
EHI Home Total Score	1.31 ± 1.35	1.35 ± 1.39
EHI Work Total Score	2.69 ± 1.26	1.05 ± 1.39
Impact Group-*n* (%)		
High	18 (50.0%)	12 (60.0%)
Low	18 (50.0%)	8 (40.0%)

^1^ Beck Depression Inventory-II; ^2^ Post Traumatic Stress Disorder Checklist, Diagnostic and Statistical Manual of Mental Disorders, 5th Edition; ^3^ Environmental Hurricane Impact Scale

**Table 3 ijerph-17-01050-t003:** Predictors of depression, anxiety, and number of medical symptoms outcome 3 to 8 weeks post hurricane landfall (Time 1).

	Beck Depression Inventory Total Score		Post Traumatic Checklist-5 Total Score		Total Number of Medical Symptoms	
	Coefficient (SE)	*p* Value	Coefficient (SE)	*p* Value	Coefficient (SE)	*p* Value
**Model 1:**						
Exposure High vs. low	5.111111	0.036 *	11.22222	0.01 *	5.111111	0.002 *
**Model 2:**						
Exposure High vs. low	5.228696	0.04 *	10.97848	0.06	4.906668	0.005 *
Past exposure	5.228696	0.27	−0.0248143	0.97	.0126636	0.57
Age	−0.051819	0.25	−0.1153924	0.47	0.0126636	0.83

* *p*-value < 0.05.

**Table 4 ijerph-17-01050-t004:** Unadjusted and adjusted Generalized Estimating Equation (GEE) models assessing change in BDI and COPE disengagement scores across the study period.

	Beck Depression Inventory Score				COPE Disengagement Score			
	Unadjusted		Adjusted		Unadjusted		Adjusted	
	Estimate (95% CI)	*p* Value	Estimate (95% CI)	*p* Value	Estimate (95% CI)	*p* Value	Estimate (95% CI)	*p* Value
**Intercept**	6.83 (4.45, 9.22)	<0.001 *	2.73 (−3.29, 8.75)	0.374	6.83 (4.45, 9.22)	<0.001	11.1 (8.75, 13.5)	<0.001 *
**Time Period**								
Time 2 vs. Time 1	0.97 (−1.62, 3.56)	0.464	**3.91 (0.24, 7.58)**	**0.037 ***	0.97 (−1.62, 3.56)	0.464	**1.75 (0.40, 3.10)**	**0.011 ***
**EHI Scores**								
Home Score Total			1.10 (−0.26, 2.45)	0.113			0.30 (−0.22, 0.81)	0.260
Work Score Total			**1.58 (0.44, 2.73)**	**0.007 ***			**0.59 (0.02, 1.16)**	**0.044 ***
**Age Group**								
≥55 vs. <55 Years			−2.55 (−5.81, 0.71)	0.123			−0.46 (– 2.32, 1.40)	0.627
**Gender**								
Female vs. Male			1.78 (−1.00, 4.56)	0.210			0.48 (−1.37, −2.32)	0.613
**Marital Status (vs. Single)**								
Married or Partnered			−2.32 (−8.32, 3.69)	0.704			**−2.21 (−4.05, −0.37)**	**0.018 ***
Divorced Widowed Separated			−2.25 (−13.8, 9.33)	0.449			−1.80 (−6.28, 2.68)	0.431

* *p*-value < 0.05. Bold text denotes significance.

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
