# Peer review of "The Short- and Long-Term Impacts of Hurricane Irma on Florida Agricultural Leaders as Early Emergency Responders: The Importance of Workplace Stability"

_ijerph, 2020, doi:10.3390/ijerph17031050_

Round 1
Reviewer 1 Report
Well written article. I suggest a few changes as indicated below with subsequent publication after accepted changes.
There are a lot of people involved in post disaster incidents A good follow-up research project would be with the emergency management impact on other personnel involved in hurricanes and tornadoes emergencies.
LINE No.
46- Instead of using as red, use in red.
47, 72- The phrase "due to" refers to dates and money. Better to use "because of" or the "result of".
62- non-professional emergency responders and public health officials are mentioned: it would be good to provide examples of these.
71 - not appropriate to start a sentence with an acronym
79 - Purpose of Study should be a sub-heading.
71 - 17 of the 27 red counties; explain briefly why.
94, 96 - percent symbol is used in charts and graphs, but in text, should be written. In lines 133-135, the symbol is acceptable.
Table 1 should have a b brief summary after it which will also help separate it from Table 2. All tables should be introduce followed by a brief summary of findings (APA).
Results sub-heading. First two paragraphs appear to be a good a good narrative on results and should precede Discussion section.
203 - Why not a section of recommendations here?
233, 239, 242 - Hodge reference needs date of publication other than access date. There are several references where the reader is left with the assumption that that the accessed date is also the date of authorship. More than likely, they are different.m. If no date is indicated in the reference, then (n. d.) is an appropriate substitution.
Reviewer 2 Report
An interesting study on occupational health in non-professional emergency responders after a natural disaster.
Occupational health status is important in all employees in securing productivity and job satisfaction.
The authors mention that similar studies have been conducted in professional emergency responders but they do not report any specific data or whether the similar measuring questionnaires/scales/tests were used. The same holds for other studies in non-professional responders. It would be interesting to see some more concrete information to help the readers assess the magnitude of impact from this study and perhaps compare results.
Also, a significant number of respondents was lost to follow up which is to be expected to a certain degree. Have the authors checked the profile of the EA's who did not respond or have they noted the reason for their loss from the study? This should be mentioned in the methods section as it could be a source of bias. Similarly, there is a possibility of information/recall bias from the study participants in providing data that should be mentioned in the limitations section.
Has job duration and prior experience in emergency response been accounted for in the sample? Is there any information about prior training received or participation in emergency response exercises from the study subjects? Some comments on these issues would add significant information to the article.
